# Relationship between Chemical Composition and In Vitro Methane Production of High Andean Grasses

**DOI:** 10.3390/ani12182348

**Published:** 2022-09-08

**Authors:** Liz Beatriz Chino Velasquez, Isabel Cristina Molina-Botero, Juan Elmer Moscoso Muñoz, Carlos Gómez Bravo

**Affiliations:** 1Escuela de Posgrado, Faculty of Animal Science, Universidad Nacional Agraria La Molina, Av. La Molina s/n-La Molina, Lima 15024, Peru; 2Faculty of Agricultural Sciences and Animal Science, Universidad Nacional de San Antonio Abad del Cusco, Av. De la Cultura 733, Cusco 08000, Peru

**Keywords:** forages, gas production, nutritional quality, ruminants

## Abstract

**Simple Summary:**

High Andean grasses have phenological cycles that are influenced by the season of the year (rainy and dry), which could affect their nutritional chemical composition and methane production. Based on this, the in vitro digestibility technique was used to measure this effect. The results of this study show that there is an effect of the chemical composition on methane production and that it changes depending on the season of the year.

**Abstract:**

The present study aims to establish the relationship between chemical composition and in vitro methane (CH_4_) production of high Andean grasses. For this purpose, eight species were collected in dry and rainy seasons: *Alchemilla pinnata*, *Distichia muscoides*, *Carex ecuadorica*, *Hipochoeris taraxacoides*, *Mulhenbergia fastigiata*, *Mulhenbergia peruviana*, *Stipa brachiphylla* and *Stipa mucronata*. They were chemically analyzed and incubated under an in vitro system. Species such as *A. pinnata* and *H. taraxacoides* were characterized by high crude protein (CP. 124 g/kg DM) and low neutral detergent fiber (NDF. 293 g/kg DM) contents in both seasons, contrary to *Stipa* grasses. This same pattern was obtained for *H. taraxacoides*, which presented the highest values of gas production, organic matter digestibility (DOM), metabolizable energy (ME) and CH_4_ production (241 mL/g DM, 59% DOM, 8.4 MJ ME/kg DM and 37.7 mL CH_4_/g DM, on average). For most species, the content of CP, acid detergent fiber (FDA) and ME was higher in the rainy season than in the dry season, which was the opposite for CH_4_ production (*p* ≥ 0.05). In general, the nutritional content that most explained the behavior of CH_4_ production was the NDF content (R^2^ = 0.69). Grasses characterized by high NDF content produced less CH_4_ (R = −0.85).

## 1. Introduction

Alpaca, vicuña and cattle production systems in the high Andean zone depend mostly on grazing perennial grasses, sedges and rosaceae [1,2]. However, these ecosystems have a high vulnerability to climate change [2], since forage productivity and chemical composition of grasses are highly dependent on environmental factors such as temperature or rainfall [3]. Likewise, there are negative effects of climate change on the intake of native grasses by ruminants, which indirectly translate into lower digestibility and greater energy loss in the rumen fermentation process, while increasing greenhouse gas emissions [4]. For example, in tropical regions, the highest emissions are associated with ruminants consuming diets with protein contents below 7% and structural carbohydrates above 70% [5].

In Peru, according to Instituto Nacional de Estadística e Información (INEI for its acronym in Spanish) [6], greenhouse gas emissions from the agriculture sector were around 26 million Gg of carbon dioxide (CO_2_) equivalent (15.2% of the national total), which are mainly concentrated in three sources: enteric fermentation, agricultural soils and manure management (41.2, 46.8 and 5.06%, respectively of emissions of agriculture sector). Methane (CH_4_) is the second most important greenhouse gas, due to the time it remains in the atmosphere (9–15 years) and its heat retention power, which is between 86 and 28 times greater than carbon dioxide over a time horizon of 20 and 100 years, respectively [7].

In the rumen of cattle, microorganisms such as bacteria, protozoa and fungi hydrolyze plant nutritional compounds to produce products such as acetate, propionate, butyrate, CO_2_ and hydrogen, among others. These last two compounds are utilized by archaea to produce CH_4_ [8]. Changes in the chemical composition of the forage can affect the amount of CH_4_ produced in the rumen [9,10]. For example, an increase in the digestibility of a feed increases the proportion of propionic acid in the total rumen fermentation products formed from it and decreases the formation of hydrogen (H_2_) and CH_4_ per unit carbohydrate fermented in the rumen [11].

The methods most commonly used by researchers to quantify methane production are the calorimetry technique (open, closed or breathing chambers), the tracer gas technique (SF_6_, nitrous oxide and CO_2_), GreenFeed™ Emissions Monitoring System and in vitro estimation [12]. This last technique consists of fermenting the feed with natural rumen microorganisms in the laboratory; this technique has the disadvantage that it does not simulate the total digestion of the animal [13], but it offers advantages such as control over the conditions of the bottle, allowing the desired number of treatments, having a low economic cost, and few requirements in terms of facilities and specialized resources. The correlation between the values obtained with in vitro digestibility and animal tests are high.

Due to the limited information available on gas emissions from rumen fermentation of typical Andean grasses and taking advantage of the multiple advantages offered by the in vitro technique, the present study aimed to measure CH_4_ production in vitro and evaluate its relationship with the chemical composition of eight grasses commonly consumed by ruminants living in the high Andean zone at two times of the year.

## 2. Materials and Methods

### 2.1. Forage Selection

The species collected in this research are natural grasses consumed by ruminants in the Peruvian Andes. The forages evaluated were: *Alchemilla pinnata* (Ruiz & Pav.), *Distichia muscoides* (Nees & Meyen), *Carex ecuadorica* (Kuekenthal), *Hipochaeris taraxacoides* (Ball), *Muhlenbergia fastigiata* (J. Presl) Henrard, *Muhlenbergia peruviana* (P.Beauv.) Steud., *Stipa brachiphylla* (Hitchc) and *Stipa mucronata* (Kunth). These were collected at two different times of the year: rainy season (February and March) and dry season (August), at the Experimental Center “La Raya”, which belongs to Universidad Nacional de San Antonio Abad del Cusco (UNSAAC, acronym in Spanish), located in the District of Marangani, Province of Canchis, Department of Cusco at an average altitude of 4313 m.a.s.l. In the dry season, rainfall ranged between 0.03 and 0.06 inches, while in February and March the amount of rainfall was between 0.54 and 2.41 inches, and the ambient temperature was 3.8 and 7.4 °C for the period with low and high rainfall, respectively; these data were collected from the meteorological station in the work area. In addition, this area has an average relative humidity of 75% and its soils are characterized by being acidic [14].

To obtain a representative sample of each of the 8 species, a pool of approximately 20 cuts was made in 10 different sites in the grazing area of the animals. In the case of the high stratum species (*S. brachiphylla*, *S. mucronata*), the cut was made from the upper part of the plants (2–3 cm), in the low stratum species (*H. taraxacoides*, *A. pinnata*, *C. ecuadorica*, *M. fastigiata* and *M. peruviana*) the cut was made at ground level and for prostrate stratum species (*D. muscoides)* the cut was made at ground level. The sampling was done with the help of a pick, removing a portion of the leaves plus the stems and then proceeding to separate the cylindrical leaves, since this is the only edible part. The forages had an average of 60 days of regrowth.

### 2.2. Evaluation of Chemical Composition

Forage samples were taken to the Animal Nutrition Laboratory of UNSAAC and the Ruminant Nutrition Laboratory of the Universidad Nacional Agraria La Molina (UNALM, acronym in Spanish) for bromatological analysis according to the methodology proposed by the Association of Analytical Chemists (AOAC) in 2005 [15]. The determination of dry matter (DM) was calculated by the difference in humidity found in the samples treated in an oven at 105 °C for 6 h (method 950.46). Crude protein (CP) content was determined by the Kjeldahl method (method 984.13) and ash by incineration in a muffle at 550 °C for 7 h (method 942.05). Fiber in neutral detergent (NDF) and acid (ADF) was determined according to method N°6 and N°5 with the filter bags technique according to Van Soest et al. [16] in the Ankom Fiber Analyzer AN 200 (Ankom^®^ Technology Corp., Macedon, NY, USA). Acid Detergent Lignin (ADL) was quantified by digestion in the DAISY II^®^ incubator (Ankom^®^ Technology Corp., NY, USA). described in method N°9. The crude fat (CF) content for the sample pool by species was analyzed using a near infrared spectrometer (NIRS. DA 7250, PerkinElmer Inc^®^., Waltham, MA, USA), in a range of the electromagnetic spectrum from approximately 700 to 2500 nanometers.

### 2.3. Gas Production

In vitro incubation of the fodder was performed for 24 h following the protocol 25.1 designed by the University of Hohenheim in Stuttgart, Germany. Initially, approximately 200 mg were placed in glass syringes of 100 mL volume (36 mm external diameter and 200 mm length). Ruminal fluid was obtained from two fistulated Jersey cows, which was filtered and transported to the laboratory for constant carbon dioxide (CO_2_) flow and magnetic stirring. Then, 20 mL of ruminal liquor mixture plus 40 mL of buffer solution (Menke and Steingass’ technique [17]) were added to the syringes, which were always kept at a temperature of 39 °C in a water bath. The incubation process was carried out in duplicate in two different incubations for each sample and time. In addition to the forage samples, a “blank” was incubated containing only the rumen liquor and medium mixture, plus two standards (ground hay and concentrate) mixed with the rumen liquor and medium. These standards have a known gas production (49.16 and 61.13 mL/200 mg DM, respectively), and a variability greater than 5% was not accepted.

### 2.4. In Vitro Methane Quantification

The methane (CH_4_) content obtained from the gas production system described above was analyzed after quantification of the gas volume through a voltmeter analyzer. This voltmeter was calibrated with molecular nitrogen (zero gas) for approximately 10 min, and purged with the calibration gas (methane standard, 12.1%). The CH_4_ concentration was obtained by directly injecting all of the gas contained in the syringes into the infrared CH_4_ analyzer (Pronova Analysentechnik GmbH and Co. KG, Berlin, Germany). The CH_4_ produced by each sample was corrected by the amount of CH_4_ produced by the syringes called “blank”.

### 2.5. Estimation of In Vitro Digestibility and Metabolizable Energy

Digestibility of organic matter (DOM, g/kg) was estimated by the formula suggested by Menke and Steingass [17] using the results of the gas production test with the Hohenheim gas test (HGT) together with CP and ash data. The formula is:(1)$${\mathrm{DOM}}_{\left(g/\mathrm{Kg}\;\right)}=149+\left(8.89\times \mathrm{GP}\right)+\left(0.448\times \mathrm{CP}\right)+\left(0.65\times \mathrm{Ash}\right)$$
where DOM is the digestibility of organic matter (g/kg); GP is in vitro gas production per 200 g DM sample (mL); CP is crude protein, N × 6.25 (g/kg); Ash is ash content (g/kg).

Metabolizable energy (ME, MJ/kg) was also estimated through the formula suggested by Menke and Steingass [17] with the results of the gas production test, CP and GP. The formula is:(2)$${\mathrm{ME}}_{\left(\mathrm{MJ}/\mathrm{kg}\right)}=1.06+\left(0.157\times \mathrm{GP}\right)+\left(0.0084\times \mathrm{CP}\right)+\left(0.022\times \mathrm{CF}\right)-0.0081$$
where ME is metabolizable energy (MJ/kg); GP is in vitro gas production per 200 g DM of sample (mL); CP is crude protein, N × 6.25 (g/kg); CF is crude fat.

### 2.6. Statistical Analysis

The SAS statistical package (version 9.2, SAS Institute, Cary, NC, USA) [18] was used to determine the effect of forage species and season on gas production, methane and the values found for metabolizable energy and in vitro digestibility of organic matter. Data were analyzed as a completely randomized design, in which each treatment (or grasses) had 4 replicates per season (rainy or dry). Means were compared using Tukey’s test (*p* < 0.05) with the model described below:
*Y_ijk_ = μ + δ_i_ + ῑ_k_ + β_ji_ +* (*δ* × *ῑ*)*_ik_ +*
*ҽ_ijk_*
(3)

where *Y_ijk_* is the response of the *j^-^th* repetition of species *i*, during season *k*; *μ* is the population mean; *δ_i_* is the effect of the *i^-^**th* species (*i* = 1...8); *ῑ_k_* is the effect of the *k^-^th* season (*k* = 1 or 2); (*δ * ῑ) ik* is the interaction between the *i^-^th* species and the *k^-^th* season; *β_ji_* is the effect of the *j^-^th* repetition per species (*j* = 1...4).

To determine the correlations between the above variables, type II linear regressions were carried out in the Origin^®^ program (Version 2016. OriginLab Corporation, Northampton, MA, USA) [19]. The linear regression model employed was:
*Y_i_ = β*_0_*+ β*_1_ × *x_i_ +*
*ҽ_i_*
(4)

where *Y_i_* is the observation of the *i^-^th* response variable, corresponding to the *i^-^th* value *x_i_* of the predictive variable *x*; *β*_0_ and *β*_1_ are the regression parameters; *x_i_* is the independent variable; and *ҽ* is the experimental error of the *i^-^th* unit.

## 3. Results

### 3.1. Chemical Composition

The bromatological analyses of the natural high Andean grasses at both times of the year are shown in Table 1. The protein content of the grasses collected ranged between 32 and 126.5 g/kg DM; the *Stipa* species was characterized by a low content of CP, while *A. pinnata* and *H. taraxacoides* provided the highest contents of CP in the alpaca diet. On average, 20 g more protein was obtained in the rainy season than in the dry season; however, species such as *H. taraxacoides* and *M. peruviana* had greater differences.

The lowest NDF content was made by the species *A. pinnata* and *H. taraxacoides* with values around 283 and 335 g/kg DM, respectively; the rest of the forage species are above 550 g/kg DM in both seasons of the year. Regarding the ADF parameter, it was observed that five (*A. pinnata*, *D. muscoides*, *C. ecuadorica*, *M. fastigiata*, and *S. mucronata*) of the eight species evaluated presented higher values in the rainy season than in the dry season (383 vs. 265 g/kg DM, respectively); the other three grasses evaluated showed minimal differences. *S. brachiphylla* and *S. mucronata* grasses had the highest NDF and ADF contents. Like CP, the species *A. pinnata* and *H. taraxacoides* have the highest ADL values, which is approximately 3 times higher than the content made by *S. brachiphylla* (101 vs. 29 g/kg DM). This tendency is similar for both times of the year. Likewise, *C. ecuadorica* is part of the group with the lowest ADL content. The ash content was higher in the species *A. pinnata*, followed by species such as *H. taraxacoides*, *C. ecuadorica* and *M. fastigiata.*

### 3.2. Gas Production, In Vitro Digestibility, Metabolizable Energy and Enteric Methane Production

The results obtained from the in vitro incubation of the grass species are shown in Table 2. For the net gas production parameter, the species *H. taraxacoides* produced the highest amount of gas in both seasons (≥235 mL/g DM), followed by species such as *C. ecuadorica* and *A. pinnata*, while *D. muscoides*, *S. mucronata* and *M. fastigiata* were among the species with the lowest gas production (89 mL/g DM on average) (*p* ≤ 0001). Of the five species that showed differences between seasons, four obtained the highest values in the dry season (*p* ≤ 0001). Adding to the above, an effect of season on the differences obtained between treatments was observed (*p* ≤ 0001).

When the results of in vitro digestibility of organic matter (DOM) were compared, it was observed that, as well as gas production, the species *H. taraxacoides* degrades on average 26% more than species *M. fastigiata*, *M. peruviana*, *S brachiphylla* and *S. mucronata* (59.7 vs. 33.7%, on average, *p* ≤ 0001). The season of the year did not affect the digestible content of the organic matter (*p =* 0.150).

Calculated metabolizable energy ranged from 3.87 to 8.47 MJ/kg DM, both values obtained in dry season for *H. taraxacoides* and *S. mucronata* species, respectively (*p* ≤ 0001). The only two species that showed a season effect were *M. fastigiata* and *A. pinnata*; however, the former species (*M. fastigiata*) contained 1.66 MJ ME/kg DM more in the rainy season than in the dry season, contrary to the behavior obtained with the species *A. pinnata*, which provided 1.99 MJ ME/kg DM more in the dry season (*p* ≤ 0.05).

Regarding CH_4_ gas production, it was observed that this variable was affected by the harvesting season (*p* ≤ 0001). In the rainy season, the maximum values were obtained with the species *H. taraxacoides*, which is 2.5 times more than *M. fastigiata* (36.4 vs. 14.8 mL/g DM); however, this difference is somewhat reduced for the dry season (1.6 times, *p* ≤ 0.05). The highest CH_4_ production was obtained in all species in the grass samples collected during the dry season, except in *M. peruviana* where there is no difference. On average, 4 mL CH_4_/g DM more is produced in the dry season than in the rainy season (*p* ≤ 0.05).

When the relationship between CH_4_ gas production and DOM was calculated, it was observed that at both times of the year the species that emitted the most CH_4_ was *H. taraxacoides*, followed by *C. ecuadorica* and *A. pinnata* (see Figure 1, *p* ≤ 0.05); for the rest of the species their production ranged between 4.68 and 7.69 mL CH_4_/g DOM. This variable was affected by the season, since in the rainy season the forages produced between 0.5 and 5.27 mL CH_4_/g DOM more than in the dry season (*p* ≤ 0.05).

### 3.3. Relationship between Chemical Composition and In Vitro Methane Production

The relation between CH_4_ production from different chemical components of pasture through in vitro gas production can be seen in Figure 2 and Table 3. Methane production is moderately explained by compounds such as CP, ADF and ADL (R^2^ = 0.27, on average), while the variations in the production of this gas are strongly explained by the NDF content (R^2^ = 0.69). In addition to the above, the contents of NDF and ADF have a negative connection between medium and high (R = −0.85 and −0.57, respectively; *p* ≤ 0.05) production of CH_4_ gas, contrary to the correlations with ADL, which were moderately positive (R = 0.55. *p* ≤ 0.05). The correction between the variables of CP and CH_4_ production were not significant.

## 4. Discussion

The productivity of the animals that inhabit the Peruvian highlands is directly related to the nutritional quality of the forage [20]. According to Febres et al. [21], native pasture plants such as *Festuca rigescens*, *Calamagrostis amoena*, *Hipochoeris taraxacoides* and *Stipa brachychylla*, among others, are of low quality, as they contribute approximately 62.0, 579 and 392 g/kg of CP, NDF and ADF, respectively, to the diet of cattle and camelids. However, these nutritional contents can be negatively affected during summer when their availability is low [22]. This coincides with the results obtained in the present study, where most grasses presented higher protein and NDF values during the rainy season, or with the results found by Alvarado–Bolovich et al. [23], who state that native grasses in the Andes increased their protein value by 34% during the rainy season (110 vs. 72 g/kg DM) though showing only a 3% increase their NDF value during the rainy season (677 vs. 660 g/kg DM). This can be explained by the fact that perennial species grow faster due to the greater availability of nutrients in the soil during the rainy season [24]. In addition, factors such as temperature and humidity in the environment directly affect plant growth and metabolism, which is why an increase in temperature, which normally occurs during the rainy season, causes the reserve carbohydrates of grasses to be reduced and increases compounds such as cellulose, lignin and pentoses, as well as increasing the percentage of total nitrogen and soluble nitrogen (Bernal [25]). This is in agreement with the data reported for CP, lignin and NDF for 6 of the 8 species evaluated.

The values of the structural wall components obtained in grasses are similar to those reported by other authors under the same altiplano environmental conditions. For example, Mamani-Linares et al. [26] report values of 684.6 g NDF/kg DM and 331.1 g ADF/kg DM for *M. fastigiata,* and 677.5 g NDF and 387.8 g ADF/kg DM for *M. peruviana* in the Chilean altiplano. Likewise, Rodriguez et al. [27] report NDF and ADF data for the dry season (730 and 740 g/kg DM) for the grass *M. fastigiata.*

Compounds such as protein and fiber are good indicators of the nutritional quality of a feed; for example, at the rumen level, the nitrogen content in the diet plays a very important role in the microbial activity. According to Tedeschi et al. [28], an intake of less than 7% protein in the diet can restrict the activity of *fibrolytic* bacteria, thus reducing the digestibility of organic matter and gas production. This may explain what happened with grasses such as *Stipa mucronata*, *Stipa brachiphylla* and *Muhlenbergia peruviana*. Contrary to this, protein values above 7% are considered to enhance microbial multiplication in the rumen, thus improving fermentation. This is clearly observed in this study with the grass *H. taraxacoides*, which was characterized by a good protein content plus a low fiber content; therefore, the forage was degraded more rapidly by the bacteria, as evidenced by the high values of gas production.

As previously indicated, degradability is inversely related to the amount of NDF present in the feed, as corroborated by several in vitro studies in which different types of grasses, legumes or tropical fruits have been incubated [5,8,29,30]. It is also indirectly correlated with the time of year, this is demonstrated by Salazar [31], in a study carried out in both seasons of the year with the species *Alchemilla pinnata* and *Hipochaeris taraxacoides*, where DOM in the dry season was 66 and 59%, while in the rainy season this value increased by between 2 and 14%. However, in the present study the relationship between NDF and DOM in the two seasons of the year is not clear, perhaps because DOM was not directly quantified but was estimated through compounds such as GP, CP and ash.

Based on the estimated values of DMD and digestible energy (DE), the data could indicate an energy deficiency if ruminants only feed during the whole year on species such as *D. muscoides*, *M. peruviana*, *M. fastigiata*, *S. brachiphylla* and *S. mucronata*, since their digestibility is very low (33% approximately); this brings with it nutritional problems that influence productive and reproductive parameters.

In recent years, studies have been conducted in Peru to measure enteric CH_4_ emissions with camelids or cattle [21,23,32,33] and although their results vary, all authors agree on the relationship between CH_4_ emissions and nutritional compounds such as fiber, which can be significantly affected in the dry season [34]. These in vivo results are not different from those obtained in this experiment conducted under controlled conditions, where 7 of the 8 species evaluated showed higher CH_4_ production during the dry season. Likewise, a medium to high negative correlation was reported between NDF and CH_4_ production, i.e., the higher the content of this compound, the lower the methane concentration. In the present investigation, the high NDF content in forages makes the degradation of fiber by bacteria slower and, therefore, the production of compounds such as dihydrogen and carbon dioxide, necessary for the formation of CH_4_, are lower. Likewise, it is observed that NDF is a useful variable for predicting methane; this same conclusion was reached by Ellis et al. after obtaining an R^2^ between these two variables of 0.63 [35].

## 5. Conclusions

The results obtained in this research show that the nutritional content of high Andean grasses varies according to environmental conditions such as rainfall and temperature in the area. In the great majority of these forages the highest contents of CP, NDF, ADF, Lignin and Ash were obtained during the rainy season. Species such as *A. pinnata* and *H. taraxacoides* were characterized by high crude protein and low neutral detergent fiber (NDF) content in both seasons, in contrast to *Stipa* sp. Grasses.

The variation in nutritional content according to the season of the year led to differences in total gas production, metabolizable energy content, and methane production. In vitro CH_4_ production was higher in the dry season than in the rainy season, and in both seasons, it was led by the species *H. taraxacoides*.

The nutritional content that most explained the behavior of methane production of forages in the Peruvian Andes was the NDF content (R^2^ = 0.69). Grasses characterized by high NDF content produced less CH_4_ (R = −0.85 between NDF and CH_4_). This same tendency was observed when related to acid detergent-treatment fiber (ADF) (R = −0.57), but this was positive when related to acid detergent-treatment lignin (ADL) content (R = 0.55). The high NDF and ADL content in some of the forage species reduced the digestibility of organic matter by ruminal microorganisms and, therefore, reduced net gas production and methane concentration during incubation.

## Figures and Tables

**Figure 1 animals-12-02348-f001:**
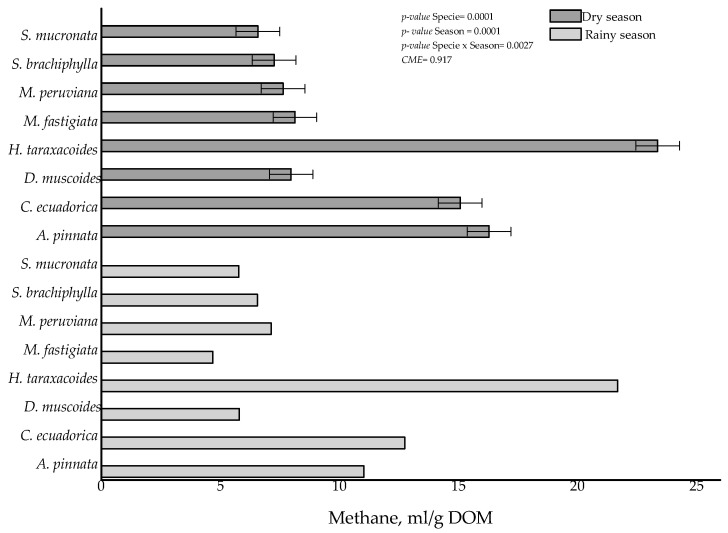
Methane production (mL) per gram of degraded organic matter (DOM) of high Andean grasses in the rainy and dry seasons.

**Figure 2 animals-12-02348-f002:**
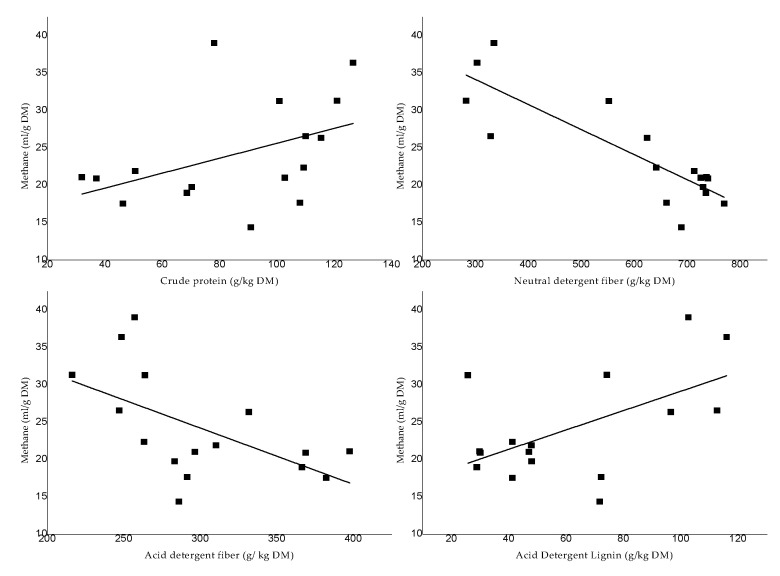
Relation between chemical composition and methane production.

**Table 1 animals-12-02348-t001:** Chemical composition of natural grasses in the rainy and dry season.

Chemical Composition (g/kg DM)	Season	Grasses							
		*A. pinnata*	*C. ecuadorica*	*D. muscoides*	*H. taraxacoides*	*M. fastigiata*	*M. peruviana*	*S. brachiphylla*	*S. mucronata*
Crude protein	Rainy	109.98	115.35	108.03	126.51	90.93	102.77	68.63	46.40
	Dry	120.98	100.81	109.26	78.14	70.36	50.62	32.08	37.03
Neutral detergent Fiber	Rainy	328.45	623.39	672.68	303.10	684.67	727.32	732.28	772.14
	Dry	283.10	549.90	637.11	334.55	730.30	718.31	733.47	738.33
Acid detergent Fiber	Rainy	246.90	331.53	291.34	248.40	286.06	297.30	368.67	383.40
	Dry	144.08	265.10	263.26	256.99	283.90	310.75	397.63	368.81
Acid Detergent Lignin	Rainy	112.57	96.50	72.14	115.90	71.60	46.55	29.15	41.21
	Dry	74.70	25.78	41.21	102.59	47.97	47.84	29.37	30.33
Ash	Rainy	246.61	165.75	60.93	168.71	134.27	59.57	63.16	58.88
	Dry	153.34	72.02	68.56	127.55	48.39	61.51	63.10	65.81
Crude fat	Rainy	1.580	1.240	1.480	2.860	2.040	1.720	1.390	1.070
	Dry	1.510	0.875	2.100	2.760	2.080	2.250	1.120	1.160

Abbreviations: Dry matter (DM).

**Table 2 animals-12-02348-t002:** Gas production, in vitro digestibility, metabolizable energy and enteric methane production of natural grasses in rainy and dry seasons.

Item	Season	Grasses								SEM	*p*-Value		
		*A. pinnata*	*C. ecuadorica*	*D. muscoides*	*H. taraxacoides*	*M. fastigiata*	*M. peruviana*	*S brachiphylla*	*S. mucronata*		Species	Season	Species × Season
Gas production (mL/g DM)	Rainy	126.32 ^cB^	172.64 ^b^	94.56 ^efB^	235.08 ^aB^	83.00 ^fB^	107.38 ^d^	101.05 ^de^	94.47 ^efA^	5.442	0.0001	0.0001	0.0001
	Dry	194.06 ^bA^	180.19 ^c^	109.83 ^dA^	248.86 ^aA^	105.38 ^dA^	106.85 ^d^	105.31 ^d^	83.67 ^eB^				
Organic Matter Digestibility (%)	Rainy	41.48 ^c^	48.40 ^b^	32.81 ^d^	59.56 ^a^	32.59 ^d^	33.97 ^d^	34.60 ^d^	32.95 ^d^	5.677	0.0001	0.1500	0.0024
	Dry	52.02 ^ab^	48.20 ^bc^	35.65 ^c^	59.86 ^a^	41.20 ^cd^	34.85 ^d^	34.42 ^d^	31.47 ^d^				
Metabolizable energy (MJ/kg DM)	Rainy	5.16 ^bB^	6.51 ^b^	4.67 ^d^	8.39 ^a^	5.59 ^dA^	4.42 ^d^	4.45 ^d^	4.15 ^d^	0.684	0.0001	0.005	0.0067
	Dry	7.15 ^bA^	6.70 ^b^	4.24 ^d^	8.47 ^a^	3.98 ^dB^	4.49 ^d^	4.36 ^d^	3.87 ^d^				
Methane (mL/g DM)	Rainy	26.58 ^bB^	26.35 ^bB^	17.66 ^dB^	36.42 ^aB^	14.38 ^fB^	21.00 ^c^	18.96 ^cdB^	17.53 ^dB^	2.369	0.0001	0.0001	0.0024
	Dry	31.32 ^bA^	31.28 ^bA^	22.37 ^cA^	39.05 ^aA^	19.74 ^cA^	21.91 ^c^	21.08 ^cA^	20.89 ^cA^				

Abbreviations: DM: Dry matter. Capital letters indicate differences within species and between seasons. Lowercase letters indicate differences within and between seasons and treatments. SEM: mean square error.

**Table 3 animals-12-02348-t003:** Correlation between nutritional content and methane production.

Relation	Equation	R^2^	SE Slope	SE Intercept	R	*p-*Value
CP, g/kg DM (*x*) on CH_4_, mL/g DM (*y*)	Y = 0.095x + 15.95	0.119	0.055	4.984	0.423	0.103
NDF, g/kg DM (*x*) on CH_4_, mL/g DM (*y*)	Y = −0.03x + 44.20	0.690	0.0056	3.499	−0.847	0.0001
ADF, g/kg DM (*x*) on CH_4_, mL/g DM (*y*)	Y = −0.07x + 46.53	0.277	0.028	8.738	−0.571	0.021
ADL, g/kg DM (*x*) on CH_4_, mL/g DM (*y*)	Y = 0.1x + 16.46	0.254	0.051	3.471	0.551	0.027

Abbreviations: CP: Crude protein, NDF: Neutral detergent fiber, ADF: Acid detergent fiber, ADL: Acid detergent lignin, CH_4_: Methane. R = Correlation coefficient, R^2^ = Coefficient of determination, SE = Standard error.

## Data Availability

All authors ensure that all data and materials support the findings and comply with field standards. The data presented in this study are available on request from the corresponding author.

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
