# Peer review of "Relationship between Chemical Composition and In Vitro Methane Production of High Andean Grasses"

_animals, 2022, doi:10.3390/ani12182348_

Round 1
Reviewer 1 Report
General comments:
The authors presented a study about the measurement of CH4 emissions in vitro and evaluate their relationship with the chemical composition of grasses commonly consumed by ruminants living in the high Andean zone at 89 two times a year. The subject is interesting, but the manuscript quality is poor and an important revision is essential. The manuscript is inappropriate for publication in this Journal under its present form.
To help the authors in the eventual revision of the manuscript, some relevant comments are listed below:
- Add Sections of Nomenclature and Abbreviations.
- Improve the format of numbers.
- Add figures about the founded correlation.
- Delete the axes secondary to the figure presented in the job.
- Delete points from titles and subtitles.
- Add a logic diagram.
- Specific comments:
Introduction
- Improve this section.
3. Results
3.2. Net gas and methane production, digestibility and metabolizable energy
- You must use the English language.
3.3. Relationship between chemical composition and in vitro methane production
- What is x?, is the CH4 production? How do you define CH4 production as a function of ME and BMD and only one variable appears in the correlation? BMD and ME are they dependent or not? Both variables depend on others that are independent... Please, revise and clarify.
Conclusion
- Improve this section.
Author Response
Dear Reviewer,
We thank the editor and reviewers for the suggestions. In this document, we describe the changes made.
Please see the attachment.

Reviewer 2 Report
Relationship between chemical composition and in vitro methane production of high Andean grasses
This study focus on the evaluation of lesser-known grasses. It follows standard methodologies and seems to be properly executed. However there are several issued which need to be addressed.
simple summary:
the last section reads very similar to the abstract (but without values)
please rephrased to make this section simpler or more accesible to a public not in academia
introduction (final section)
I feel a paragraph o sentence is missing to connect this sentence and the objective
currently, it reads somehow odd, authors state the disadvantages of in vitro methods, yet, they continue with the objective where they employ in vitro methods
a possibility would be to present first the disadvantages and then their advantages and then continue with the objective
Methodology
There are several issues to be addressed (mark in file), the main comments are:
I understand the possible correction by the amount of methane in the "blank"
However, I do not understand the rationale of "correcting" by two standards.
can you provide the formula or rationale employed for such corrections?
It is not totally clear why the author chooses to predict or estimate DOM instead of measuring it.
As predicted DOM (as well as ME) arose from GP and chemical composition, then, the factors (for example, GP, CP, ash, CF) employed in their prediction would now be correlated or associated and could no longer be used as predictors.
Statistical analysis
in methodology, authors declare only two incubation per season?
where the six replicates per season come from?
Results
there are minor issues mark in the file
a general comment is related to the fact that the authors seems to manage as synonyms "chemical composition" and "nutritional contribution". Text must be revised and proper terminology used
conclusion
the more digestible, the more ME and also likely more methane
what it is really important is to state the relationship in terms of methane in relation to digestibility (mlCH4/gDOM) or as % of GE or ME
otherwise, the final message seems to be that a plant with the poorest nutritional value would be best
see further comment in the file

Author Response
Dear Editor
We thank you for your comments and suggestions, in the attached file you will find the changes made.

Round 2
Reviewer 1 Report
The work has been substantially improved, so its publication is recommended.
Author Response
Best regards
We thank the editor and reviewers for the suggestions. In this document, we describe the changes made.

Reviewer 2 Report
Relationship between chemical composition and in vitro methane production of high Andean grasses
The author made a great effort to include and answer many of the comments raised in the first review round.
However, important aspects remain to be solved.
Detailed comments can be check on the revised file, however must of the comments are related with the following aspects.
.- Author must carefully check their tabular data when presenting and discussion their results.
for example -
Authors declare in the abstract, discussion and conclusion "In general, the lowest CH4 production occurred when the NDF content in the grass was low (R2≥0.69)." However, according to figure 2 and the equation presented, it is the opposite the higher NDF content, the lower the CH4 production.
Also, in results, they declare that ME content was higher in rainy season when in fact table 2 shows that most of the species have higher ME content during the Dry season
Table 3 (correlation results) was missing, however author do refer to correlation results using regression table. Seems to mix up R and R^2. it is advised to properly check and correct the information provided (see file for further comments)
Regarding the statistical analysis, results and discussion. Please take in account the following aspect.
Authors are comparing season and species effects only and are ignoring the significant interaction effects
A significant interactions indicate that not all responses followed the same trend (for example for some species, the ME value is higher in dry season while for other species the higher values was found during the rainy season, the same can be said for other variables).
authors should made and effort to explain the significan differences denoted by the interaction. Why some species perform different?
Given the previous comment, author must avoid generalizations (some are mark in the main text - see file) as the species behave differently
For example (lines 314-315) "since the reduced diet quality observed in the dry period is accompanied by a low crude protein content and an increase in cell wall constituents." this is not true for all species, according to the chemical composition results, some species do have higher CP and lower NDF during the dry season.
Author must make an effort to carefully revise their results and improve the text.

Author Response

(The authors gave the same response as above.)

Round 3
Reviewer 2 Report
Relationship between chemical composition and in vitro methane production of high Andean grasses
The authors have properly answered comment and questions from the previous review round
I do not have further comments